# Impact of COVID-19 Pandemic on School-Aged Children’s Physical Activity, Screen Time, and Sleep in Hong Kong: A Cross-Sectional Repeated Measures Study

**DOI:** 10.3390/ijerph191710539

**Published:** 2022-08-24

**Authors:** Hung-Kwan So, Gilbert T. Chua, Ka-Man Yip, Keith T. S. Tung, Rosa S. Wong, Lobo H. T. Louie, Winnie W. Y. Tso, Ian C. K. Wong, Jason C. Yam, Mike Y. W. Kwan, Kui-Kai Lau, Judy K. W. Kong, Wilfred H. S. Wong, Patrick Ip

**Affiliations:** 1Department of Paediatrics and Adolescent Medicine, Li Ka Shing Faculty of Medicine, The University of Hong Kong, Hong Kong, China; 2Department of Health and Physical Education, The Education University of Hong Kong, Hong Kong, China; 3Department of Pharmacology and Pharmacy, Li Ka Shing Faculty of Medicine, The University of Hong Kong, Hong Kong, China; 4Department of Ophthalmology and Visual Sciences, The Chinese University of Hong Kong, Hong Kong, China; 5Department of Paediatrics and Adolescent Medicine, Princess Margaret Hospital, Hong Kong, China; 6Department of Medicine, Li Ka Shing Faculty of Medicine, The University of Hong Kong, Hong Kong, China; 7Inspiring HK Sports Foundation, Hong Kong, China

**Keywords:** COVID-19 pandemic, school closures, school-aged children, physical activity, screen time, sleep, Hong Kong

## Abstract

Despite concerns about the negative effects of social distancing and prolonged school closures on children’s lifestyle and physical activity (PA) during the COVID-19 pandemic, robust evidence is lacking on the impact of the pandemic-related school closures and social distancing on children’s wellbeing and daily life. This study aimed to examine changes in the PA levels, sleep patterns, and screen time of school-aged children during the different phases of the COVID-19 outbreak in Hong Kong using a repeated cross-sectional design. School students (grades 1 to 12) were asked to report their daily electronic device usage and to fill in a sleep diary, recording their daily sleep onset and wake-up time. They were equipped with a PA monitor, Actigraph wGT3X-BT, to obtain objective data on their PA levels and sleep patterns. Students were recruited before the pandemic (September 2019–January 2020; *n* = 577), during school closures (March 2020–April 2020; *n* = 146), and after schools partially reopened (October 2020–July 2021; *n* = 227). Our results indicated lower PA levels, longer sleep duration, and longer screen time among participants recruited during school closures than those recruited before the COVID-19 outbreak. Primary school students were found to sleep on average for an extra hour during school closures. The later sleep onset and increased screen time documented during school closures persisted when schools partially reopened. Our findings illustrate the significant impact of social distancing policies during the COVID-19 pandemic on the sleep pattern, screen time, and PA level in school-aged children in Hong Kong. Professionals should urgently reinforce the importance of improving physically activity, good sleep hygiene, and regulated use of electronic devices for parents and school-aged children during this unprecedented time.

## 1. Introduction

During the coronavirus disease 2019 (COVID-19) pandemic, social distancing policies such as home confinement, school closures, and minimal/no outdoor play for school children, resulted in unusual lifestyle changes in children and adolescents [1]. It has been proposed that the lockdown measures have affected the general well-being of children and adolescents more profoundly than the infection itself [2]. Since the outbreak, there has been increasing interest in the impact of these policies on physical activity (PA), screen time, and sleep patterns among children and adolescents. According to a study in Shanghai, the prevalence of physically inactive school-aged children increased from 21% in January 2020 to 66% in March 2020 [3]. However, countries such as Colombia, Spain, Italy, and Germany observed an increase in PA during the pandemic [4,5], whereas five studies found no significant differences in PA behavior before and during the pandemic [5,6,7]. The status, intensity, and duration of COVID-19 restrictions have varied by country and region and may explain the inter-regional differences in children’s PA during the pandemic [8]. A recent report in the US showed that the non-academic screen time of children doubled during the COVID-19 pandemic from 3.8 to 7.7 h a day [9]. This raises concerns regarding the known potential harms of excessive screen time, mainly the risks to physical and mental health such as greater obesity and higher depressive symptoms, among school-aged children [10]. In addition, evidence showed that school-aged children had a longer sleep duration, with delayed sleep onset and wake-up times, as a result of homeschooling during this period [8,11,12]. Although adequate sleep is beneficial for children, researchers found a pooled prevalence of sleep disturbances of up to 54% in children during the pandemic [13]. Some researchers have suggested that the COVID-19 pandemic could have exacerbated existing lifestyle problems such as physical inactivity and sedentary lifestyles due to missed PA classes and social distancing measures [14,15].

However, the majority of evidence in the current literature is limited; most studies have been based on subjective questionnaire assessments and only looked at the short-term changes across the first year of the pandemic. It would be meaningful to examine the longer-term impact of the pandemic on the lifestyles of children and adolescents. Although the association between lifestyle and well-being in school-aged children has been widely studied, research using objective assessments for parameters such as physical activity and sleep patterns is needed to document quantitative evidence on the lifestyle changes during the COVID-19 pandemic. To the best of our knowledge, only one study has used Actigraph data to evaluate sleep patterns in 16 healthy Japanese preschoolers during the pandemic [16]. In our study on school-aged children in Hong Kong, we objectively measured and compared lifestyle habits, including PA levels, sleep patterns, and screen time, during the different phases of the COVID-19 outbreak in order to examine the impact of policy on children.

## 2. Materials and Methods

This was a repeated cross-sectional study [17] designed to assess the impact of COVID-19 on school-aged children in Hong Kong over 2 years across 3 time-points, including pre-pandemic (September 2019–January 2020), school closures (March 2020–April 2020), and partial school reopening (October 2020–July 2021). We explored changes in PA levels, sleep patterns, and electronic device usage (screen time) in primary and secondary school students. Repeated cross-sectional data assessed independent samples in each time period, allowing for meaningful inference of changes across a population during the different time periods. Although overlap of cross-sectional samples is advantageous because it minimizes the variance of parameter estimates, considerable sample overlap is not required for reliable conclusions regarding population trends over time [18].

### 2.1. Participants

The study sample was divided into the five main districts of Hong Kong: Hong Kong Island, West Kowloon, East Kowloon, West New Territories, and East New Territories. Using a stratified random sampling method, a list of primary (grades 1–6) and secondary (grades 7–12) schools from each district was randomly generated. The primary and secondary schools in each district were invited to participate according to the list. If the invitation was rejected, the next school on the list was invited until the required sample size was met. One grade of students was randomly selected from each school to participate. This method was adopted in our previous local prevalence study published in a peer-reviewed journal [19].

The first assessment was conducted between September 2019 and January 2020 before the first wave of the COVID-19 outbreak in Hong Kong. Schools that agreed to participate in the study were contacted by mail and telephone. Written informed consent was obtained from the participants’ parents. Each student and parent pair received a set of questionnaires and a wrist-placed activity monitor (Actigraph wGT3X-BT, Pensacola, FL, USA). The Actigraph device is known for its sensitivity and has been used to monitor PA and sleep in children in several peer-reviewed publications [20,21,22,23].

The second assessment was conducted between late February and March 2020, approximately two months after school closures due to the first wave of COVID-19 in Hong Kong. Parents willing to participate in the assessments were contacted by telephone. A set of questionnaires and an Actigraph device were delivered to each participating household. To comply with the social-distancing policies and to minimize infection risk, the consent form, questionnaire, and Actigraph device were left by the door of each household. The signed consent forms, questionnaires, and PA monitoring device were collected at the end of the monitoring period.

The third assessment was conducted between October 2020 and July 2021 after schools were partially reopened. Students were required to attend half-day in-person classes at school and then half-day online lessons at home. Participating schools were contacted by mail and phone. After the signed consent form was returned, the questionnaire and Actigraph set was distributed to students and parents.

### 2.2. Measurements

#### 2.2.1. Questionnaires

The questionnaires were completed by one parent of the student. Demographic information and children’s electronic device usage (screen time) data were collected [24]. Participants were also asked to fill in a sleep diary daily to record their sleep onset and wake-up times.

#### 2.2.2. Screen Time

Screen time was assessed by asking participants how many hours on average they spent using an electronic device (e.g., TV, online/electronic game, smartphone, computer, tablet) on a weekday and at the weekend. The electronic device usage was calculated using the weighted average formula: [(weekday × 5) + (weekend × 2)]/7.

#### 2.2.3. Actigraph Data

To properly analyze physical activity, subjects were required to have three valid monitoring days. One day’s recording with more than 10 h wear time beyond the sleep period was counted as a valid day [25]. Subjects with data from at least two days and nights were used in the sleep analysis.

##### Non-Wear Time

The non-wear period was identified using the Choi algorithm [26]. A minimum of 90 consecutive “zero” epochs with a spike tolerance of 2 min and 100 counts/min were used to identify non-wear periods.

##### Sleep Measurements

Bedtime was determined by five consecutive asleep minutes whereas wake time was determined by ten consecutive awake minutes. Sleep periods beyond the sleep time and wake-up time marked in the students’ sleep diary or over 50% non-wear time were eliminated.

Sleep Period Classification

Extracted data was analyzed using 60-s (1 min) epochs. Each 1-min epoch was classified as asleep or awake using the Cole–Kripke algorithm. The Cole–Kripke algorithm adjusts the epoch data by rescaling the count values by 100 and setting values over 300 to 300 [27]. A sleep index less than 1 was classified as asleep and otherwise as awake [28]. Sleep periods were detected using the Tudor–Locke algorithm [25].

Sleep Parameters

The measurement of sleep-related parameters was based on previously described studies [21,29]. Parameters included sleep and wake-up time, total sleep duration, sleep latency, sleep efficiency, and sleep fragmentation index (SFI). Sleep efficiency was calculated as the percentage of asleep time divided by the total in-bed time. The SFI indicates restlessness during the sleep period [30] and was calculated as a sum of the movement index and the fragmentation index. The movement index was the percentage of total awake minutes out of the total in-bed time and the fragmentation index was the percentage of one-minute periods of sleep over the entire sleep period. Latency was defined as the difference between the students’ self-reported sleep time and accelerometer-detected sleep time.

Physical Activity Levels

Step count and sum of the vector magnitude (VM) in 60-s epochs were processed using ActiLife (software version 6.13.4, ActiGraph, FL, USA). Cut-offs for PA and sedentary behavior according to Chandler et al. were used to quantify the time spent as sedentary (VM < 3672 per min), engaged in light PA (3672 ≤ VM per min < 9816), moderate PA (9816 ≤ VM per min < 23,628), and moderate-to-vigorous PA (MVPA) (VM per min ≥ 23,628) [31].

### 2.3. Statistical Analysis

All analyses were conducted using R statistical software, version 4.0.3 (R Program for Statistical Computing, Lucent Technologies, NJ, USA). Principal component analysis was used to construct the socioeconomic status (SES) index using household income, parental education level, and occupation. Descriptive analyses were used to compare the demographic characteristics of the students among the three periods. A mixed effect model was used to compare the measurement variables of lifestyle (physical activity, sleep, and electronic device usage) among the three periods, adjusting for age, sex, and SES status, for the whole sample and separately for primary and secondary schools.

Compositional data analysis (CoDA) was conducted using “compositions” and “robCompositions” r packages to explore the relationship between the daily use of time (sleep, MVPA, screen time, and other activities) and sleep quality of children among the three periods. Small non-zero values were used in the imputation to replace zeros, as log ratios cannot be created with zero values [32]. Compositional isotemporal substitution analyses were performed to explore the influence of the reallocation of fixed time durations (15 min) between the daily time use of different activities on children’s sleep quality using SFI as the dependent variable. All models were adjusted for sex and SES status. Statistical significance was set at a two-sided *p* < 0.05 for all analyses.

### 2.4. Ethics

This study was approved by The Institutional Review Board of the University of Hong Kong/Hospital Authority Hong Kong West Cluster (IRB number: UW 19-516). Informed written consent was obtained from the parents of the participants.

## 3. Results

### 3.1. Descriptive Analysis (Table 1)

A total of 577 students (267 primary and 310 secondary students) were recruited before the pandemic (September 2019–January 2020), 146 students (87 primary and 59 secondary students) were recruited during school closures (March 2020–April 2020), and 293 students (137 primary and 156 secondary students) were recruited after schools partially reopened (October 2020–July 2021). Table 1 shows a summary of the participant baseline characteristics. There were no significant differences in students’ anthropometric parameters or SES index among the three periods. The proportion of primary school students recruited during school closures was higher (60% during school closures vs. 46% before the pandemic and 47% after school reopened, *p* = 0.013). The percentage of female students recruited during school closures and reopening was higher (68% and 77% during school closures and reopening, respectively, vs. 58% in the pre-pandemic phase, *p* < 0.001). Students recruited during the pre-pandemic period were slightly older than those in the other two periods (12.85 years vs. 12.14 years during school closures, *p* = 0.006 and 11.93 years after schools partially reopened, *p* < 0.001).

**Table 1 ijerph-19-10539-t001:** Characteristics of the participants at three time-points.

	Pre-Pandemic	During School Closure	School Partially Reopened	
**Data collection period**	September 2019–January 2020	March 2020–April 2020	October 2020–July 2021	*p*-value
	N (%)/Mean (SD)	N (%)/Mean (SD)	N (%)/Mean (SD)	
**Total valid data collected**	577	146	293	
**School type**				
**Primary**	267 (46%)	87 (60%)	137 (47%)	**0.013**
**Secondary**	310 (54%)	59 (40%)	156 (53%)	
**Sex**				
**Female**	333 (58%)	100 (68%)	227 (77%)	**<0.001**
**Male**	244 (42%)	46 (32%)	66 (23%)	
**Age (years)**	12.85 (2.61)	12.14 (2.90)	11.93 (2.11)	**<0.001**
**Anthropometric parameters**				
**Height (z-score)**	−0.39 (1.50)	−0.41 (1.49)	−0.29 (1.10)	0.535
**Weight (z-score)**	−0.10 (1.32)	−0.03 (1.38)	−0.02 (1.25)	0.672
**BMI (z-score)**	0.17 (1.25)	0.24 (1.26)	0.13 (1.29)	0.683
**Body fat (%)**	20.99 (9.63)	21.38 (9.27)	20.72 (8.67)	0.788
**Body Status**				
**Underweight**	12 (2%)	2 (1%)	12 (4%)	0.576
**Normal**	442 (77%)	111 (76%)	227 (77%)	
**Overweight**	85 (15%)	21 (14%)	38 (13%)	
**Obese**	34 (6%)	7 (5%)	15 (5%)	
**Missing**	4 (1%)	5 (3%)	1 (0%)	
**Social Economic Status (SES)**	−0.08 (1.53)	0.18 (1.69)	0.03 (1.52)	0.301

### 3.2. Changes in PA, Sleep Time, and Screen Time (Table 2)

After adjusting for age, sex, and SES index, the average step count and MVPA time per day decreased during school closures and stayed lower after schools’ partial reopening compared to the pre-pandemic time. Moreover, the average sedentary time per day increased when schools reopened.

Compared to their counterparts prior to the pandemic, students had lower sleep efficiency and higher SFI though with a longer sleep duration during schools’ closure. Students had a longer sleep latency and lower sleep efficiency after schools’ reopening.

Students’ screen time for both academic and non-academic purposes increased during both school closures and after schools’ reopening compared to the pre-pandemic time.

Comparing between SES, a lower SES was associated with higher SFI, lower MVPA time, and longer total screen time pre-pandemic and during school closures. Lower SES was also associated with a longer sedentary time during school closures.

**Table 2 ijerph-19-10539-t002:** Changes in weekly PA, sleep, and screen time.

	Pre-Pandemic (Reference)	During School Closures			Schools Partially Reopened		
	Mean (SD)	Mean (SD)	β (SE)	*p*-Value	Mean (SD)	β (SE)	*p*-Value
**Physical activity**							
**Step count per day**	10,969.33 (2492.15)	8472.51 (3295.39)	−2692.83 (235.04)	**<0.001**	9547.04 (2455.05)	−1562.17 (189.24)	**<0.001**
**Sedentary time per day (h)**	10.46 (1.45)	10.64 (1.83)	0.16 (0.16)	0.292	10.85 (1.69)	0.61 (0.12)	**<0.001**
**MVPA time per day (h)**	0.47 (0.35)	0.42 (0.49)	−0.09 (0.03)	**0.007**	0.44 (0.33)	−0.08 (0.03)	**<0.001**
**Sleep**							
**Latency (min)**	43.66 (50.83)	55.61 (81.31)	7.56 (6.48)	0.243	56.83 (52.13)	11.96 (4.72)	**0.011**
**Actual sleep duration (h)**	7.45 (1.03)	8.76 (1.28)	1.19 (0.11)	**<0.001**	7.55 (1.09)	−0.07 (0.09)	0.438
**Sleep time (HH:MM)**	23:29 (00:53)	00:30 (01:17)	62.43 (5.68)	**<0.001**	23:39 (00:51)	14.92 (4.52)	**0.001**
**Wake time (HH:MM)**	07:11 (00:39)	09:54 (01:31)	115.14 (6.26)	**<0.001**	07:25 (00:59)	12.53 (4.98)	**0.012**
**Efficiency (%)**	95.72 (4.18)	91.07 (8.7)	−4.27 (0.54)	**<0.001**	95.25 (4.21)	−0.86 (0.43)	**0.047**
**SFI**	27.62 (7.34)	35.3 (11.75)	7.91 (0.88)	**<0.001**	27.56 (7.345)	0.72 (0.7)	0.306
**Screen time**							
**Level of parental guidance**	1.68 (0.56)	1.65 (0.56)	−0.05 (0.06)	0.407	1.76 (0.54)	0.03 (0.04)	0.497
**Non-academic screen time per day (h)**	4.38 (4.3)	4.54 (4.55)	1.16 (0.4)	**0.003**	5.60 (4.22)	1.87 (0.32)	**<0.001**
**Academic screen time per day (h)**	1.01 (1.33)	1.69 (2.14)	0.6 (0.17)	**<0.001**	2.92 (2.24)	1.93 (0.14)	**<0.001**
**Overall screen time per day (h)**	5.39 (5.10)	6.23 (5.80)	1.79 (0.48)	**<0.001**	8.52 (5.40)	3.85 (0.38)	**<0.001**

MVPA—Moderate to vigorous physical activity.

### 3.3. Changes in PA, SLEEP Time, and Screen Time in Primary and Secondary School Students

#### 3.3.1. Primary Schools (Grade 1 to Grade 6) (Table 3)

After adjusting for age, sex, and SES index, the average step count and MVPA time per day decreased during school closures compared with the pre-pandemic time. Students had a longer sleep latency, higher SFI, and increased academic screen time per day during school closures.

After schools partially reopened, compared to the pre-pandemic time, students had lesser step counts per day, shorter sleep duration, poorer sleep efficiency during school days, and an increase in both academic and non-academic screen time in both school days and holidays.

**Table 3 ijerph-19-10539-t003:** Changes in weekly PA, sleep, and screen time among primary school students.

	Pre-Pandemic (Reference)	During School Closures			Schools Partially Reopened		
	Mean (SD)	Mean (SD)	β (SE)	*p*-Value	Mean (SD)	β (SE)	*p*-Value
**Physical activity**							
**Step count per day**	11,902.36 (2100.06)	9594.29 (2445.65)	−1961.05 (312.71)	**<0.001**	10,433.02 (2493.44)	−1165.94 (258.07)	**<0.001**
**School days**	11,999.24 (2026.46)	-	-	-	10,682.54 (2663.60)	−1127.8 (256.02)	**<0.001**
**Holidays**	11,713.80 (3599.98)	9594.29 (2445.65)	−1960.99 (471.21)	**<0.001**	9836.68 (3248.60)	−1317.95 (411.12)	**0.001**
**Sedentary time per day (h)**	10.12 (1.29)	10.07 (1.61)	−0.25 (0.19)	0.181	10.13 (1.46)	0.00 (0.15)	0.977
**School days**	10.23 (1.36)	-	-	-	10.33 (1.51)	0.21 (0.16)	0.202
**Holidays**	9.91 (1.80)	10.07 (1.61)	0.03 (0.24)	0.899	9.56 (2.08)	−0.59 (0.21)	**0.005**
**MVPA time per day (h)**	0.64 (0.33)	0.50 (0.33)	−0.11 (0.05)	**0.018**	0.62 (0.35)	0.02 (0.04)	0.572
**School days**	0.63 (0.33)	-	-	-	0.61 (0.35)	0.01 (0.04)	0.881
**Holidays**	0.68 (0.51)	0.50 (0.33)	−0.17 (0.07)	**0.016**	0.65 (0.49)	0.05 (0.06)	0.473
**Sleep**							
**Latency (min)**	53.19 (40.88)	83.68 (64.82)	27.65 (6.78)	**<0.001**	60.31 (44.30)	5.77 (5.54)	0.298
**School days**	53.96 (44.17)	-	-	-	62.87 (42.91)	8.58 (5.52)	0.120
**Holidays**	50.41 (59.49)	83.68 (64.82)	31.29 (9.13)	**0.001**	52.69 (62.63)	−0.64 (8.08)	0.937
**Actual sleep duration (h)**	7.95 (0.79)	8.80 (1.03)	0.85 (0.13)	**<0.001**	7.77 (0.90)	−0.17 (0.1)	0.089
**School days**	7.72 (0.79)	-	-	-	7.44 (0.83)	−0.27 (0.1)	**0.007**
**Holidays**	8.66 (1.47)	8.80 (1.03)	0.11 (0.21)	0.589	8.85 (1.37)	0.08 (0.18)	0.658
**Sleep time (HH:MM)**	23:07 (01:21)	00:15 (01:22)	64.3 (7.09)	**<0.001**	23:21 (00:45)	13.25 (5.79)	**0.022**
**School days**	22:57 (00:45)	-	-	-	23:11 (00:45)	13.41 (5.53)	**0.015**
**Holidays**	23:37 (00:54)	00:15 (01:22)	35.39 (9.16)	**<0.001**	23:52 (01:11)	10.77 (8.11)	0.184
**Wake time (HH:MM)**	07:16 (00:36)	09:22 (01:15)	119.96 (6.33)	**<0.001**	07:21 (00:39)	6.37 (5.17)	0.218
**School days**	06:49 (00:26)	-	-	-	06:51 (00:29)	2.95 (3.41)	0.388
**Holidays**	08:43 (01:37)	09:22 (01:15)	35.78 (13.60)	0.009	09:00 (01:26)	14.83 (12.04)	0.218
**Efficiency (%)**	95.81 (3.53)	94.13 (4.45)	−1.15 (0.50)	0.023	95.92 (3.35)	−0.19 (0.41)	0.639
**School days**	97.13 (2.94)	-	-	-	96.43 (3.66)	−0.80 (0.40)	**0.044**
**Holidays**	91.61 (8.82)	94.13 (4.45)	2.47 (1.08)	**0.022**	94.28 (4.85)	1.93 (0.95)	**0.043**
**SFI**	26.89 (6.57)	31.11 (7.87)	4.01 (1.00)	**<0.001**	27.23 (6.70)	0.41 (0.82)	0.614
**School days**	25.44 (6.73)	-	-	-	26.85 (6.97)	1.24 (0.87)	0.153
**Holidays**	31.87 (12.06)	31.11 (7.87)	−0.24 (1.54)	0.874	28.97 (8.95)	−2.68 (1.37)	0.050
**Screen time**							
**Level of parental guidance**	1.87 (0.51)	1.94 (0.49)	0.04 (0.08)	0.586	1.86 (0.52)	0.02 (0.06)	0.765
**Non-academic screen time per day (h)**	1.81 (2.69)	2.65 (3.60)	0.56 (0.45)	0.211	4.5 (3.78)	2.67 (0.37)	**<0.001**
**School days**	1.39 (2.42)	-	-	-	3.87 (3.76)	2.27 (0.34)	**<0.001**
**Holidays**	2.86 (4.09)	2.65 (3.60)	0.95 (0.66)	0.153	6.09 (5.27)	3.62 (0.54)	**<0.001**
**Academic screen time per day (h)**	0.38 (0.92)	0.88 (1.28)	0.55 (0.21)	**0.010**	2.98 (2.38)	2.44 (0.18)	**<0.001**
**School days**	0.43 (0.91)	-	-	-	3.49 (3.01)	3.02 (0.22)	**<0.001**
**Holidays**	0.54 (1.21)	0.88 (1.28)	0.7 (0.21)	**0.001**	1.71 (1.87)	0.95 (0.18)	**<0.001**
**Overall screen time per day (h)**	2.28 (3.31)	3.53 (1.65)	1.12 (0.58)	0.055	7.48 (5.32)	5.11 (0.48)	**<0.001**
**School days**	1.83 (3.02)	-	-	-	7.36 (6.02)	5.28 (0.49)	**<0.001**
**Holidays**	3.4 (4.65)	3.53 (1.65)	1.65 (0.74)	**0.027**	7.79 (5.75)	4.56 (0.61)	**<0.001**

Latency—Difference between monitor-recorded and self-reported sleep time (min); MVPA—Moderate to vigorous physical activity; SFI—Sleep fragmentation index.

#### 3.3.2. Secondary Schools (Grade 7 to Grade 12) (Table 4)

During school closure, students had decreased step counts per day, less MVPA time per day, more sedentary time, poorer sleep efficiency, and a higher SFI compared with the pre-pandemic time. Academic and non-academic screen time increased during school closures.

After schools partially reopened, students had lesser step counts, lesser MVPA, and increased sedentary time per day compared with pre-pandemic data. A longer sleep latency, shorter sleep duration, lower SFI, and increased academic plus non-academic screen time were also documented.

**Table 4 ijerph-19-10539-t004:** Changes in weekly PA, sleep, and screen time among secondary school students.

	Pre-Pandemic (Reference)	During School Closures			Schools Partially Reopened		
	Mean (SD)	Mean (SD)	β (SE)	*p*-Value	Mean (SD)	β (SE)	*p*-Value
**Physical activity**							
**Step count per day**	9692.94 (2492.86)	6111.86 (2196.68)	−4035.39 (381.47)	<0.001	8795.19 (2157.77)	−1519.01 (348.77)	**<0.001**
**School days**	9997.45 (2422.58)	-	-	-	9454.58 (2218.24)	−1333.44 (360.77)	**<0.001**
**Holidays**	8786.08 (4481.52)	6111.86 (2196.68)	−2955.5 (612.03)	<0.001	7550.07 (2575.63)	−1514.08 (641.59)	**0.018**
**Sedentary time per day (h)**	10.79 (1.48)	11.61 (1.51)	0.79 (0.27)	0.003	11.48 (1.63)	0.83 (0.23)	**<0.001**
**School days**	10.93 (1.64)	-	-	-	11.93 (1.80)	1.07 (0.27)	**<0.001**
**Holidays**	10.62 (1.82)	11.61 (1.51)	1.11 (0.31)	<0.001	10.92 (1.68)	0.41 (0.32)	0.194
**MVPA time per day (h)**	0.32 (0.27)	0.17 (0.18)	−0.15 (0.04)	0.001	0.28 (0.22)	−0.07 (0.04)	0.061
**School days**	0.31 (0.26)	-	-	-	0.27 (0.20)	−0.09 (0.04)	**0.020**
**Holidays**	0.41 (0.54)	0.17 (0.18)	−0.24 (0.07)	0.001	0.28 (0.24)	−0.09 (0.07)	0.220
**Sleep**							
**Latency (min)**	33.16 (58.25)	5.08 (84.24)	−22.8 (12.02)	0.058	53.97 (57.78)	23.03 (10.02)	**0.022**
**School days**	37.32 (59.97)	-	-	-	65.50 (61.85)	31.19 (9.86)	**0.002**
**Holidays**	10.86 (70.18)	5.08 (84.24)	0.02 (15.86)	0.999	26.71 (68.40)	10.68 (15.66)	0.495
**Actual sleep duration (h)**	6.94 (0.95)	8.72 (1.64)	1.79 (0.22)	<0.001	7.31 (1.20)	−0.01 (0.19)	0.976
**School days**	6.58 (0.92)	-	-	-	6.41 (0.99)	−0.35 (0.16)	**0.030**
**Holidays**	8.65 (1.65)	8.72 (1.64)	0.23 (0.31)	0.466	8.87 (1.41)	0.39 (0.32)	0.235
**Sleep time (HH:MM)**	23:57 (00:52)	00:54 (01:03)	51.12 (9.47)	<0.001	23:56 (00:50)	8.57 (8.63)	0.321
**School days**	23:52 (00:53)	-	-	-	23:53 (00:54)	12.73 (8.75)	0.146
**Holidays**	00:29 (01:05)	00:54 (01:03)	16.82 (12.59)	0.182	00:05 (01:05)	−21.18 (13.41)	0.114
**Wake time (HH:MM)**	07:05 (00:42)	10:45 (01:30)	207.70 (11.73)	<0.001	07:29 (01:12)	11.63 (10.53)	0.269
**School days**	06:38 (00:27)	-	-	-	06:26 (00:32)	−9.84 (4.71)	**0.037**
**Holidays**	09:35 (01:33)	10:45 (1:30)	64.15 (18.75)	0.001	09:22 (01:35)	−0.86 (19.64)	0.965
**Efficiency (%)**	95.56 (4.31)	85.48 (11.69)	−9.36 (1.08)	<0.001	94.58 (4.79)	−1.16 (0.98)	0.236
**School days**	96.03 (4.42)	-	-	-	95.73 (4.78)	−0.34 (0.81)	0.674
**Holidays**	91.95 (8.08)	85.48 (11.69)	−6.17 (1.67)	<0.001	92.27 (6.94)	−1.2 (1.75)	0.491
**SFI**	28.28 (7.69)	43.12 (13.89)	14.5 (1.62)	<0.001	27.82 (7.96)	0.03 (1.48)	0.986
**School days**	27.53 (8.06)	-	-	-	25.74 (9.17)	1.07 (0.27)	**<0.001**
**Holidays**	33.46 (11.11)	43.12 (13.89)	9.38 (2.14)	<0.001	31.14 (9.00)	−0.54 (2.30)	0.813
**Screen time**							
**Level of parental guidance**	1.45 (0.53)	1.26 (0.40)	−0.16 (0.09)	0.067	1.68 (0.54)	0.04 (0.08)	0.572
**Non-academic screen time per day (h)**	6.6 (4.19)	7.64 (4.27)	0.61 (0.77)	0.422	6.44 (4.36)	0.69 (0.64)	0.278
**School days**	5.8 (4.43)	-	-	-	5.62 (4.19)	0.39 (0.68)	0.559
**Holidays**	8.62 (5.99)	7.64 (4.27)	1.36 (1.04)	0.192	8.48 (5.6)	1.46 (0.85)	0.087
**Academic screen time per day (h)**	1.49 (1.45)	2.99 (2.59)	1.29 (0.34)	<0.001	2.87 (2.13)	1.36 (0.28)	**<0.001**
**School days**	1.49 (1.49)	-	-	-	3.01 (2.28)	1.53 (0.3)	**<0.001**
**Holidays**	1.49 (1.73)	2.99 (2.59)	1.27 (0.38)	0.001	2.53 (2.28)	1.08 (0.32)	**0.001**
**Overall screen time per day (h)**	8.09 (4.84)	10.64 (5.25)	1.91 (0.92)	0.039	9.31 (5.34)	2.12 (0.75)	**0.005**
**School days**	7.29 (5.11)	-	-	-	8.63 (5.3)	1.97 (0.81)	**0.015**
**Holidays**	10.11 (6.55)	10.64 (5.25)	2.65 (1.17)	0.023	11.01 (6.38)	2.86 (0.95)	**0.003**

Latency—Difference between monitor-recorded and self-reported sleep time (min); MVPA—Moderate to vigorous physical activity; SFI—Sleep fragmentation index.

### 3.4. Influence of the Reallocation of Time Use on Sleep Quality

Table 5 shows the results of the compositional data analysis of the ‘15-min time reallocation’ effects on sleep quality during the three periods.

**Table 5 ijerph-19-10539-t005:** Compositional data analysis of 15-min time reallocation effects on sleep quality during different periods.

		Predicted Difference (95% Confidence Interval)							
**Add 15 min**	Remove 15 min	Overall		Pre-pandemic		During school closure		After school partially reopened	
**Screen time**	MVPA	0.865	(0.563, 1.168)	0.332	(−0.177, 0.840)	**2.013**	**(0.481, 3.546)**	0.546	(−0.114, 1.206)
**Screen time**	Sleep	−0.025	(−0.063, 0.012)	−0.016	(−0.178, 0.145)	0.074	(−0.063, 0.212)	−0.018	(−0.046, 0.009)
**Screen time**	Other activities	0.013	(−0.02, 0.045)	0.001	(−0.152, 0.153)	0.099	(−0.024, 0.223)	0.011	(−0.016, 0.039)
**MVPA**	Screen time	−0.616	(−0.831, −0.401)	−0.229	(−0.834, 0.376)	**−1.293**	**(−2.212, −0.374)**	−0.343	(−0.759, 0.072)
**MVPA**	Sleep	−0.638	(−0.864, −0.413)	−0.242	(−0.608, 0.124)	**−1.171**	**(−2.118, −0.224)**	−0.362	(−0.798, 0.074)
**MVPA**	Other activities	−0.600	(−0.816, −0.384)	−0.225	(−0.569, 0.119)	**−1.146**	**(−2.092, −0.200)**	−0.332	(−0.749, 0.085)
**Sleep**	Screen time	0.022	(−0.031, 0.075)	0.012	(−0.425, 0.450)	−0.123	(−0.316, 0.070)	0.018	(−0.009, 0.046)
**Sleep**	MVPA	0.890	(0.575, 1.205)	0.348	(−0.179, 0.874)	**1.939**	**(0.372, 3.506)**	0.564	(−0.116, 1.244)
**Sleep**	Other activities	0.038	(0.024, 0.051)	0.017	(−0.008, 0.042)	0.024	(−0.032, 0.081)	0.029	(−0.008, 0.067)
**Other activities**	Screen time	−0.015	(−0.063, 0.032)	−0.004	(−0.449, 0.441)	−0.148	(−0.324, 0.029)	−0.010	(−0.034, 0.015)
**Other activities**	MVPA	0.853	(0.548, 1.158)	0.331	(−0.174, 0.836)	**1.914**	**(0.350, 3.477)**	0.536	(−0.126, 1.197)
**Other activities**	Sleep	−0.038	(−0.051, −0.024)	−0.017	(−0.042, 0.008)	−0.025	(−0.078, 0.027)	−0.028	(−0.063, 0.006)

MVPA—Moderate to vigorous physical activity.

Reallocation of MVPA time significantly affected sleep quality during school closures. As MVPA time increased, the sleep fragmentation index decreased and vice versa. There was no significant influence on sleep quality from the reallocation of time for any factors, including sleep, screen time, and MVPA during the pre-pandemic and school partial reopening phases.

## 4. Discussion

To the best of our knowledge, this is one of the first studies using objective PA and sleep quality monitoring data to demonstrate changes in sleep patterns and PA levels among school-aged children during different phases of the COVID-19 pandemic. Previous studies conducted in other countries relied on self-reported questionnaires and compared their findings with pre-COVID data from other studies, which introduced potential bias [15,33,34,35]. Our study provides robust quantitative data using a repeated cross-sectional design and technology to record data to compare pre-pandemic, school closure, and school partial reopening periods. We objectively demonstrated that children had lower PA levels, longer sleep duration, and longer screen time during school closures (March 2020–April 2020) compared with the pre-pandemic time (September 2019–January 2020). Although some of these parameters improved after schools’ reopening (October 2020 and July 2021), the impacts of physical inactivity, prolonged screen time, and sleep disturbances appeared to persist. This study found that the COVID-19 pandemic is having potentially significant long-term impacts on the lifestyle and physical activity of school-aged children, which adds an important piece of evidence to the current literature.

In Hong Kong, only around 8% of school-aged children fulfill the recommended PA levels set by the World Health Organization in 2013 [36]. Our study found that physical inactivity deteriorated after the COVID-19 outbreak compared to pre-pandemic levels. Given the strong link between childhood physical inactivity and long-term major diseases such as metabolic syndrome, obesity, and cardiovascular problems, urgent attention is required. The promotion of physical activity, encouragement of such environments in schools, and education of childcare providers, parents, and teachers to check for potential lifestyle changes in children during the pandemic [37,38] should be prioritized. Promoting out-of-school PA by guiding students, helping them to set realistic goals, and encouraging them to self-monitor would have long-term benefits [39]. In a recent study conducted by Tso et al., children and their families had more psychosocial problems during the COVID-19 pandemic, with more significant difficulties reported in vulnerable families such as in children with chronic illnesses and those with special educational needs. In particular, the quality of life of these children was found to be negatively correlated with physical inactivity and prolonged screen time [40]. Findings from our study imply that the worsening physical inactivity during prolonged school closures persisted after the reopening of schools, which may pose further long-term health risks to children and future adults.

There were significant changes in sleep patterns during the different study periods. Although sleep duration was longer during school closures than the pre-pandemic time, this was still within the recommendations of the American Academy of Sleep Medicine [41]. However, the overall sleep quality was poorer as evidenced by the delayed bedtime and wake-up time, increased in-bed duration, longer sleep latency, and increased movement and SFI. During school closures, some primary school students reported sleeping after 2 a.m., which was not observed in the pre-pandemic group. These children likely have a disturbed circadian rhythm. The circadian rhythm is an endogenous rhythm that assists humans in keeping their biological clock on a 24-h cycle. [42]. In children and adolescents, circadian rhythm disorders and sleep problems are known to be associated with poorer cognitive performance; behavioral problems such as hyperactivity-impulsivity, inattention, daytime sleepiness, emotional and dysregulation; and physical problems such as obesity [43,44,45,46]. In the present study, replacing 15 min/day of screen time, sleep duration, or other activities with MVPA reduced SFI significantly during school closures. During the school closure period, all students attended online courses from home instead of going to school. Without proper parental supervision, the increase in screen time could cause later sleep and wake-up times, resulting in an overall longer sleep duration. Moreover, although most of the sleep patterns recovered after schools partially reopened, a slight delay in bed time was still found. A delayed sleep phase may result in mental health challenges such as lower learning capacity/academic performance, addiction behaviors, and elevated levels of anxiety and depression. Parental and community education and involvement to ensure good sleep habits in children is vital [47,48].

Excessive screen time was also reported in primary school students. The switch to online teaching meant longer use of electronic devices for academic purposes, but recreational screen time increased concurrently and remained elevated when schools partially reopened compared to the pre-pandemic time [49]. Excessive screen time in school-aged children has been associated with an increased risk of behavioral issues, obesity, cardio-metabolic problems, and myopia [50,51,52]. The American Academy of Pediatrics recommends that families and pediatricians should collaborate to develop a family media use plan to guide their children on media use [53].

The findings of this study need to be interpreted with the following caveats. Firstly, the sample size was relatively small due to the design of objective PA monitoring during the COVID-19 outbreak. Secondly, the study subjects were recruited on a voluntary basis, which suggests selection bias may be a concern. However, the availability of repeated measurements for the lifestyle behaviors, over nearly 18 months, allowed us to assess the associations between the behaviors alone and in combination with the three time periods (pre-pandemic, school closure, and school partial reopening). Our study was able to dynamically capture previously unreported differences in children’s PA levels and sleep patterns using an objective PA monitoring approach.

## 5. Conclusions

This study found that there were adverse changes in sleep patterns, screen time, and physical activity levels during the period of school closures, which persisted when schools partially reopened. As Hong Kong is now in the fifth wave of COVID-19, since early January 2022, the Hong Kong Government has again tightened social distancing measures, which may further increase the long-term risks for children. The improvement in sleep quality with a 15-min increase in MVPA is worth further attention. Further study is needed to longitudinally follow up these children and their families. Professionals should provide parents and children with appropriate advice on maintaining physical activity, good sleep discipline, and appropriate use of electronic devices during this challenging period, particularly during school closures.

## Data Availability

All data that support the findings of this study are available from the corresponding author upon reasonable request.

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
