# Peer review of "Impact of COVID-19 Pandemic on School-Aged Children’s Physical Activity, Screen Time, and Sleep in Hong Kong: A Cross-Sectional Repeated Measures Study"

_ijerph, 2022, doi:10.3390/ijerph191710539_

Round 1
Reviewer 1 Report
1. Why only one grade randomly selected ?
2. The sample size in tbe 3 phases are totally different so how they are comparable
3. The sample are different in each phase so how we can measure the effect of covid ?
4. Table so condensed, need to be revised
Author Response
Thank you so much for taking the time to review this manuscript and making insightful comments!
The point-to-point responses to your comments are as follows.
Reviewer 1
Comments and Suggestions for Authors
- Why only one grade randomly selected?
A: One grade was randomly selected from each school to minimize the disturbance of the schools and the workload of the teachers. We also aim to involve more schools to join this study.
- The sample size in the 3 phases are totally different so how they are comparable
A: Our study design was based on repeated cross-sectional data where a survey is administered to a new sample of subjects at 3-time points. For example, the British Social Attitudes Survey (BSA) is a repeated cross-sectional survey conducted in most years since 1983. Because repeated cross-sectional data take a different sample of a population over time, they are used for analyzing group changes over time (also known as aggregate change over time). They cannot be used to look at individual change.
Ref: https://www.bsa.natcen.ac.uk/
- The sample are different in each phase so how we can measure the effect of covid ?
A: As mentioned in the above response, we aim to determine the group changes over time. Therefore, we adjusted the results with age and gender. All participating schools were on the same school list.
- Table so condensed, need to be revised
A: Thank you for the comment. All tables have been revised.
Sincerely,
Dr Patrick Ip
Reviewer 2 Report
Current version of manuscript is globally average.
In my opinion main points of WEAKNESSES are the following:
First of all, ENGLISH LANGUAGE must be improved (i.e. sentences such as at lines 46-49 must be re-wiritten in a better form).
Moreover, quality of tables should be greatly ameliorated.
Finally, I'd like to suggest some articles in order to insert in references and/or use for enlarging discussion/etc.
PLoS One. 2021;16(7):e0255520. doi: 10.1371/journal.pone.0255520.
Prev Med. 2021 Feb;143:106349. doi: 10.1016/j.ypmed.2020.106349.
Sci Rep. 2021 Apr 20;11(1):8529. doi: 10.1038/s41598-021-88071-4.
Therefore, SOME MAJOR REVISIONs are necessary before considering acceptance by Editors.
Best regards.
Author Response
Thank you so much for taking the time to review this manuscript and making insightful comments!
The point-to-point responses to your comments are as follows.
Reviewer 2
Comments and Suggestions for Authors
Current version of manuscript is globally average.
In my opinion main points of WEAKNESSES are the following:
First of all, ENGLISH LANGUAGE must be improved (i.e. sentences such as at lines 46-49 must be re-wiritten in a better form).
A: The ENGLISH LANGUAGE has been revised by a native English-speaking colleague.
Moreover, quality of tables should be greatly ameliorated.
A: The editable form of the tables has been provided instead of the pictures. Thus the quality of the tables has been improved.
Finally, I'd like to suggest some articles in order to insert in references and/or use for enlarging discussion/etc.
A: I have added “Sci Rep. 2021 Apr 20;11(1):8529” as reference 15 and “Prev Med. 2021 Feb;143:106349” as reference 48 to enrich the introduction and discussion. I don’t include “PLoS One. 2021;16(7):e0255520” because it is a protocol for systematic review.
Sincerely,
Dr Patrick Ip
Round 2
Reviewer 2 Report
I have much appreciated efforts of authors in responding to reviewers' suggestions.
Therefore, for me modified paper is suitable for considering publication by Editors.
Best regards.